# Transcriptome Analysis for Salt-Responsive Genes in Two Different Alfalfa (*Medicago sativa* L.) Cultivars and Functional Analysis of *MsHPCA1*

**DOI:** 10.3390/plants13081073

**Published:** 2024-04-11

**Authors:** Qican Gao, Ruonan Yu, Xuesong Ma, Hada Wuriyanghan, Fang Yan

**Affiliations:** 1Key Laboratory of Forage and Endemic Crop Biology, Ministry of Education, School of Life Sciences, Inner Mongolia University, Hohhot 010070, China; 18330697642@163.com (Q.G.); yuruonan2016@sina.com (R.Y.); 18847666832@163.com (X.M.); 2Crop Cultivation and Genetic Improvement Research Center, College of Agricultural, Hulunbuir University, Hulunbuir 021008, China

**Keywords:** alfalfa, cultivars, salt stress, differentially expressed genes, H_2_O_2_, *MsHPCA1*

## Abstract

Alfalfa (*Medicago sativa* L.) is an important forage legume and soil salinization seriously affects its growth and yield. In a previous study, we identified a salt-tolerant variety ‘Gongnong NO.1’ and a salt-sensitive variety ‘Sibeide’. To unravel the molecular mechanism involved in salt stress, we conducted transcriptomic analysis on these two cultivars grown under 0 and 250 mM NaCl treatments for 0, 12, and 24 h. Totals of 336, and 548 differentially expressed genes (DEGs) in response to NaCl were, respectively, identified in the ‘Gongnong NO.1’ and ‘Sibeide’ varieties. The Kyoto Encyclopedia of Genes and Genomes (KEGG) and Gene Ontology (GO) pathway enrichment analysis showed that the DEGs were classified in carbohydrate metabolism, energy production, transcription factor, and stress-associated pathway. Expression of *MsHPCA1*, encoding a putative H_2_O_2_ receptor, was responsive to both NaCl and H_2_O_2_ treatment. MsHPCA1 was localized in cell membrane and overexpression of *MsHPCA1* in alfalfa increased salt tolerance and H_2_O_2_ content. This study will provide new gene resources for the improvement in salt tolerance in alfalfa and legume crops, which has important theoretical significance and potential application value.

## 1. Introduction

Alfalfa (*Medicago sativa* L.) is recognized as a crucial forage crop globally due to its nitrogen-fixing capacity and potential as a biofuel feedstock for ethanol production [1,2]. Alfalfa is a moderately salt-tolerant crop. Unfortunately, soil salinization has significantly increased in recent years, affecting approximately 33% of the world’s arable land [3]. This limits the biomass yield and distribution of alfalfa [4,5]. Therefore, enhancing the salt resistance of alfalfa and understanding the mechanisms of salinity tolerance are especially important.

Salinity is a crucial determinant of crop yield worldwide [6]. Exposure to salt stress leads to a decrease in yield or even the death of cultivated crops. The primary components of soil salinity are sodium chloride, magnesium, calcium sulfates, and bicarbonates. Plants have developed numerous physiological and biochemical tolerance mechanisms to cope with salt stress. These include the accumulation of compatible solutes such as glycine betaine and proline, as well as scavenging of reactive oxygen species (ROS) like singlet oxygen (O_2_), superoxide radicals (O_2_^−^), hydrogen peroxide (H_2_O_2_), and hydroxyl ions (OH^−^). Additionally, plants maintain cellular ionic equilibrium by regulating the homeostasis of ions such as Na^+^, K^+^, Ca^2+^, and Cl^−^ [7,8,9,10,11,12]. Ethylene (ET), abscisic acid (ABA), gibberellins (GAs), jasmonic acid (JA), melatonin (MT), and auxin are phytohormones that can influence plant responses to salt stress. These hormones may play key roles in regulating growth and stress responses [13,14,15,16,17]. Energy and carbon skeletons play important roles in seed germination and seedling growth [18,19]. Salt stress can result in a decrease in osmotic and turgor pressure, which progressively limits cell expansion and leaf growth. This is due to the limited availability of energy and metabolites caused by osmotic stress, ionic toxicity, and oxidative stress [20,21,22]. Seed germination is the most critical phase of seedling establishment and is highly sensitive to salt stress. In recent years, numerous transcriptional and translational analyses have identified salinity-responsive genes in the seedlings of various plants such as rice [23], Suaeda fruticose [24], maize [25], wheat [26], tomato [27], cotton [5], soybean [28], and alfalfa [29,30]. It has been established that certain salt-responsive genes, including transcription factors (TFs), plant hormones, and antioxidant capacity, play important roles in plant’s response to salt stress [31,32,33]. A comprehensive understanding of the molecular networks and the mechanisms underlying plant responses to salt stress is essential for enhancing the salinity tolerance of plants. Nevertheless, the transcriptional regulatory network involved in the salt stress response of alfalfa species, specifically between the salt-tolerant and sensitive varieties, remains poorly understood.

In the present study, we analyzed the transcriptome of alfalfa seedlings, including the salt-tolerant ‘Gongnong NO.1’ and the salt-sensitive ‘Sibeide’ varieties, following exposure to 250 mM NaCl stress for 12 and 24 h. Subsequently, we conducted a comparative analysis of gene expression in these stressed samples using RNA-Seq technology. The DEGs were found to be involved in various pathways including stress response, transcription factor, carbohydrate metabolism, and energy production. We also investigated the role of *MsHPCA1,* a putative H_2_O_2_ receptor in alfalfa, and demonstrated that *MsHPCA1* might contribute to salt tolerance in alfalfa. The results will assist in the identification of candidate genes associated with salt tolerance and a comprehensive understanding of the intricate mechanisms underlying alfalfa’s salinity tolerance.

## 2. Results

### 2.1. Physiological Responses of Alfalfa (‘Gongnong NO.1’ and ‘Sibeide’) toward NaCl Treatment

In this study, we aimed to investigate the salt response of *Medicago sativa*. We selected ‘Gongnong NO.1’ as a salt-tolerant variety and ‘Sibeide’ as a salt-sensitive variety, based on previous research [15]. During the germination test, ‘Gongnong NO.1’ showed significant inhibition under 250 mM NaCl treatment, while the ‘Sibeide’ was only moderately inhibited (Figure 1A,B). The germination rates of ‘Sibeide’ and ‘Gongnong NO.1’ alfalfa decreased by 40% and 5% under salt treatment compared with the control groups, respectively (Figure 1C). The germination potential of ‘Sibeide’ was decreased by 90–95% compared to the control group at different time points (Figure 1D). The germination potential of ‘Gongnong NO.1’ decreased by 40% after 4 days of NaCl treatment, but with prolonged treatment time, its germination potential was almost restored at 6 or 8 days compared to the control group (Figure 1D). The root growth of ‘Sibeide’ and ‘Gongnong NO.1’ was significantly retarded by 60% and 30%, respectively (Figure 1E), while shoot growth was significantly retarded by 50% and 25%, respectively (Figure 1F).

Under NaCl stress, the chlorophyll and proline content were slightly higher in ‘Gongnong NO.1’ than in ‘Sibeide’ under NaCl stress, while Malondialdehyde (MDA) content was lower in ‘Gongnong NO.1’ than in ‘Sibeide’ under salt stress (Figure 2). Taken together, the above results indicate that ‘Sibeide’ is more sensitive to salt treatment compared with ‘Gongnong NO.1’.

### 2.2. Identification of Differentially Expressed Genes in ‘Gongnong NO.1’ and ‘Sibeide’

To gain a comprehensive understanding of the transcriptome of the salt-tolerant variety ‘Gongnong NO.1’ and the salt-sensitive variety ‘Sibeide’ alfalfa seedlings, we conducted high-throughput RNA-Seq analysis after treating 7-day-old seedlings with 250 NaCl for 0 h, 12 h, and 24 h. The RNA-Seq analysis yielded between 21.85 and 23.36 million reads. The sequencing quality of the RNA-Seq tags, as assessed by saturation analysis, met the requirements for differential gene expression analyses. Approximately 80.41% of the reads were successfully mapped to either multiple (5.50%) or unique (74.91%) genomic locations, while 19.59% remained unmatched (Appendix A). Principal Component Analysis (PCA) indicated that the DEGs were expressed at background levels 12 h after salt stress in both alfalfa varieties (Figure 3A). The Venn diagram illustrated the distribution of DEGs from ‘Gongnong NO.1’ (0 h, 12 h, and 24 h) and ‘Sibeide’ (0 h, 12 h, and 24 h). A total of 6640 DEGs were expressed at 12 h and 24 h treatment stages in both alfalfa varieties; 2392 and 3030 are expressed in g12 and g24 compared with g0; while 2711 and 3403 DEGs were expressed in S12 and S24 compared with S0, respectively (Figure 3B). Furthermore, 758 DEGs were identified in ‘Sibeide’ and ‘Gongnong NO.1’ alfalfa seed-lings under normal conditions (g0 vs. S0) compared to stress conditions (g12 vs. g0, g24 vs. S0, S12 vs. S0, and S24 vs. S0,), which exhibited co-expressed DEGs between the two varieties, among them, 349 genes were up-regulated and 409 genes were down-regulated (Figure 3C). Cluster and Kyoto Encyclopedia of Genes and Genomes (KEGG) analysis revealed that the DEGs were categorized into twelve groups based on their expression patterns (Figure 3D), and associated with photosynthesis, inorganic anion transport, reactive oxygen metabolic and response to oxidative stress in g0, g12, g24, S0, S12, and S24 (Figure 3E).

### 2.3. Transcriptome Analysis of Differentially Expressed Genes in ‘Gongnong NO.1’ and ‘Sibeide’ under Salt Treatment

The DEGs between two alfalfa varieties were analyzed to identify those involved in salt stress during seed germination. A total of 1359, 824, and 1484 genes were found to be differentially regulated under salinity stress for 0 h, 12 h, and 24 h in 7-day-old germinated seedlings of the two alfalfa varieties (Figure 4A). Among these, 187 DEGs were co-expressed under normal conditions, as well as under salt-stressed conditions for 12 h and 24 h in ‘Sibeide’ and ‘Gongnong NO.1’ seedlings; 336 and 839 DEGs were, respectively, expressed under 12 h and 24 h of salt stress in both varieties (Figure 4B). The KEGG analysis categorized all the DEGs into 18 terms based on sequence homology, with the genes involved in functions such as iron ion binding, acyltransferase, defense response, response to biotic stimulus, and so on (Figure 4C). Notably, iron ion binding and acyltransferase were the dominant categories during salt stress in the seed germination of the two alfalfa varieties (Figure 4D).

### 2.4. Analysis of Differentially Expressed Genes under Salt Treatment at Different Times

#### 2.4.1. Glyco-Metabolism-Associated Gene

In ‘Gongnong NO.1’, 40 DEGs encode proteins associated with glycol-metabolism at g12 and g24 compared the g0 stage. Among these DEGs, the expression of 34 genes was up-regulated and 6 genes were down-regulated (Figure 5A). GO classifies these genes into two main categories (biological process and molecular function). These genes were associated with carbohydrate metabolic process, carbohydrate transport, UDP-glycosyltransferase activity, quercetin 3-O-glucosyltransferase activity, raffinose alpha-galactosidase activity, transferring glycosyl groups, and so on (Figure 5C). Among the DEGs, the genes are related to UDP-glucuronosyltransferase, sugar transporter, and glycosyltransferase (Appendix A). A total of 17 DEGs encoded 13 catalytic enzymes including xyloglucosyl transferase [EC:2.4.1.207], hydroquinone glucosyltransferase [EC:2.4.1.218], mannan endo-1,4-beta-mannosidase [EC:3.2.1.78], polygalacturonase [EC:3.2.1.15], chitinase [EC:3.2.1.14], beta-glucosidase [EC:3.2.1.21], cis-zeatin O-glucosyltransferase [EC:2.4.1.215], pathogen-inducible salicylic acid glucosyltransferase [EC:2.4.1.-], probable galacturonosyltransferase-like 1 [EC:2.4.1.-] raffinose synthase [EC:2.4.1.82], sucrose synthase [EC:2.4.1.13], alpha-galactosidase [EC:3.2.1.22], and alpha-L-arabinofuranosidase [EC:3.2.1.55] at g12 and g24 compared with g0 stage. The 13 DEGs encoding the above enzymes exhibited similar expression patterns with high expression levels at g12 and g24h and four genes with down expression (Figure 5E). Compared to the s0 stage, there were a total of 27 DEGs in s12 and s24 (Appendix A), of which 16 genes were up-regulated and 11 genes were down-regulated (Figure 5B). In the GO classification’s three main categories (biological process, cellular component, and molecular function), there were three, two, and four functional groups, respectively. We noticed these genes from functional groups such as glycolytic process, fructose-bisphosphate aldolase activity, transferase activity, transferring glycosyl groups, scopolin beta-glucosidase activity, beta-glucosidase activity, and so on (Figure 5D). A total of 14 DEGs encoded 11 catalytic enzymes including trehalose 6-phosphate phosphatase [EC:3.1.3.12], alpha-galactosidase [EC:3.2.1.22], glucan endo-1,3-beta-D-glucosidase [EC:3.2.1.39], galactolipid galactosyltransferase [EC:2.4.1.184], fructose-bisphosphate aldolase, class I [EC:4.1.2.13], beta-fructofuranosidase [EC:3.2.1.26], hydroquinone glucosyltransferase [EC:2.4.1.218], gallate 1-beta-glucosyltransferase [EC:2.4.1.136], polygalacturonase [EC:3.2.1.15], beta-glucosidase [EC:3.2.1.21], and xyloglucan glycosyltransferase 4 [EC:2.4.1.-] at s12 and s24 compared with the s0 stage (Figure 5F). These results demonstrated that the DEGs were involved in regulating glyco-metabolism and promoting energy catabolism during seedling growth under salt stress.

#### 2.4.2. Energy-Associated Genes

In response to salt stress, plants reduce their growth rate by reallocating energy to adapt to the adverse environment. The RNA-seq analysis revealed changes in the expression of unigenes involved in basal energy metabolism under salt stress. Two *ATPase*-*related* genes (MsG0180005882 and MsG0180000608) were up-regulated approximately 7.14~5.92- or 1.58~2.14-fold at g12 and g24 compared with the g0 stage. Four *ATPase*-*related* genes (MsG0780036265, MsG0380013942, MsG0380015758, and MsG0380016571) were up- or down-regulated by approximately 2.27~2.38- or 1.63~1.86- or −1.02~−1.09- or −1.15~−1.47-fold and three *ATP-synthase* (MsG0380017406, MsG0880046219, and MsG0880042935) were up- or down-regulated by 1.40~1.57- or 1.12~1.26- or −1.11~−1.18-fold at s12 and s24 compared with s0 stage. Six *NADPH* (*NADP*) or FAD-related *thioredoxin* genes were up- or down-regulated in ‘Gongnong NO.1’ seedlings and most *NADPH* (*NADP*) or FAD-related genes exhibited a downward trend in ‘Sibeide’ seedlings under saline stress (Appendix A).

#### 2.4.3. Transcription Factor

TFs play a crucial role in regulating seedling salt tolerance. In ‘Gongnong NO.1’ and ‘Sibeide’, 31 and 34 TFs belonging to 8 and 12 TF families, respectively, were identified (Appendix A). The most representative TF families included Zinc finger (MsG0280010800, MsG0180004800, MsG0480020428, MsG0780039229, MsG0880046674, MsG0480023441, MsG0180003673, MsG0380012512, MsG0880045175, MsG0880046752, MsG0380017253, and MsG0080048430), AP2/ERF (MsG0880042051, MsG0580024424, MsG0380017423, and MsG0180005095), WRKY (MsG0780041742 and MsG0280011473), MYB (MsG0380013522, MsG0780040741, MsG0280010653, MsG0480020725, MsG0380015227, and MsG0180004180), NAC (MsG0480023497, MsG0280009717, and MsG0180001854), and GRAS (MsG0280011041) which were mostly up-regulated in ‘Gongnong NO.1’. Conversely, in ‘Sibeide’, most TFs such as Zinc finger (MsG0580028462, MsG0180003687, MsG0880046215, MsG0480023693, MsG0080047905, MsG0180003595, MsG0880043098, MsG0580028016, MsG0580026614, MsG0580026192, MsG0380015805, MsG0580029912, and MsG0780041106), WRKY (MsG0180000525, MsG0280007786, and MsG0080048036), MYB (MsG0380014706 and MsG0580024732), GRAS (MsG0280011042, MsG0680032758, and MsG0580024328) and bHLH (MsG0580025534, MsG0880043481, and MsG0480022288) were down-regulated. The Zinc finger, MYB, WRKY, AP2/ERF, GRAS, and bHLH families contain other stress-responsive cis-regulatory elements in their promoters, directly or indirectly regulating plant salt tolerance. These differentially expressed transcription factors may contribute to the molecular mechanism underlying the difference salt resistance between the two varieties.

#### 2.4.4. Stress-Associated Gene

NaCl treatment induced the expression of stress-related pathway genes, including *peroxidase*, *glutathione-S-transferase* (*GST*), *late embryogenesis abundant* (*LEA*), and *heat shock factor* (*HSF*). ‘Gongnong NO.1’ and ‘Sibeide’, respectively, exhibited 40 and 16 identified stress-related genes (Appendix A). The up-regulation of *peroxidase*, *GST*, and *LEA* genes was observed in salt-tolerant alfalfa (‘Gongnong NO.1’), while the down-regulation of *peroxidase*, *GST* related genes was observed in salt-sensitive alfalfa (‘Sibeide’). Above all, the less inhibitory effect of saline stress on the growth of ‘Gongnong NO.1’ compared with ‘Sibeide’ may be attributed to ‘Gongnong NO.1’ having a more efficient system for salt resistance.

### 2.5. Expression Analysis of MsHPCA1

*HPCA1*, Hydrogen Peroxide-Induced Ca^2+^ Increases1, was reported to be involved in stress response and to act as a central reactive oxygen species (ROS) receptor in *Arabidopsis thaliana* [34]. Interestingly, expression of *MsHPCA1*, an alfalfa homolog of *HPCA1*, was altered upon salt treatment in the transcriptome analysis. Therefore, the role of *MsHPCA1* in alfalfa salt response was investigated in the following experiments. In order to examine the role of *MsHPCA1* in alfalfa growth and stress response, we used RT-qPCR to analyze its expression pattern. Although *MsHPCA1* mRNA was detectible in all the tissues such as root, leaf, stem, and flower, the expression pattern differed in two cultivars. The *MsHPCA1* mRNA level exhibited peak expression in the stem of ‘Gongnong NO.1’, whereas in ‘Sibeide’, peak expression was observed in the leaf. (Figure 6A,B). Abiotic stresses, drought and especially NaCl, significantly elevated *MsHPCA1* expression in ‘Gongnong NO.1’ but not in ‘Sibeide’ (Figure 6C,D). More importantly, *MsHPCA1* expression was induced by H_2_O_2_ in both ‘Sibeide’ and ‘Gongnong NO.1’, indicating its putative role as a H_2_O_2_ receptor and involvement in stress response (Figure 6C,D).

### 2.6. Subcellular Localization of MsHPCA1

Sequence analysis indicated that MsHPCA1 possessed a transmembrane domain (Figure 7A). Domain modeling also indicated that MsHPCA1 was a transmembrane receptor (Figure 7B). Evolutionary analysis indicated that MsHPCA1 belonged to HPCA1 family but was clustered into different subgroups compared with *Arabidopsis thaliana* HPCA1, a well-characterized H_2_O_2_ receptor. The MsHPCA1 protein’s tertiary structure was modeled using PDB entry I1L094.1.A as a template. The sequence similarity was 0.76, covering the range of 1 to 954 aa. And, the average similarity score was 0.99. MsHPCA1 is predicted to be a non-specific serine/threonine protein kinase (Figure 7C). Subcellular localization analysis by green fluorescent protein (GFP)-fused expression of *MsHPCA1* in the *Nicotiana Benthamiana* transient expression system showed that MsHPCA1 localized at the cell membrane, demonstrating its possible role as a membrane-localized receptor (Figure 7D).

### 2.7. Transgenic Analysis of MsHPCA1 Function

We investigated the involvement of *MsHPCA1* in salt response in alfalfa. We generated transgenic plants overexpressing *MsHPCA1* in the salt-sensitive cultivar ‘Sibeide’ and silenced *MsHPCA1* expression in the salt-resistant cultivar ‘Gongnong NO.1’. Transgenic plants were obtained by hairy root transformation, and the transgenes were verified by RFP expression as the pKGWRR vector carried the RFP gene (Figure 8A), RT-qPCR was then used to detect overexpression and silencing efficiency (Figure 8B,C).Under salt stress, the leaf of the WT plant showed a yellowing phenotype while the leaf of the *MsHPCA1*-overexpressed plant displayed a normal phenotype (Figure 8D). On the contrary, the *MsHPCA1*-silenced plant showed compromised growth compared with the WT plant under salt stress (Figure 8J). At the same time, the chlorophyll content, plant height, root length, and seedling length of transgenic plants and WT were measured for H_2_O_2_ content. The chlorophyll content, plant height, root length, and seedling length were significantly higher in plants overexpressing the *MsHPCA1* gene compared to wild type, whereas the chlorophyll content, seedling length, and root length were significantly lower in plants with suppressed *MsHPCA1* gene expression (Figure 8E–H,K–N). The *MsHPCA1*-overexpressed plant accumulated more H_2_O_2_ compared with the WT plant while the *MsHPCA1*-silenced plant showed no differences (Figure 8I,O). The results showed that *MsHPCA1* was associated with H_2_O_2_ accumulation, and overexpression of *MsHPCA1* improved the salt tolerance of plants.

## 3. Discussion

### 3.1. Physiological Responses of Seedlings in Two Different Types of Alfalfa to Salt-Stress

Through the analysis of germination rates, biomass, and physiological indicator under salt treatment, we identified ‘Gongnong NO.1’ as a salt-tolerant cultivar and ‘Sibeide’ as a salt-sensitive cultivar (Figure 1 and Figure 2), consistent with previous studies [15,35]. To gain insight into the salt tolerance mechanisms of Alfalfa, high-throughput transcriptome sequencing was conducted on two cultivars exposed to salinity. Salt stress can induce osmotic stress and oxidative stress in plants [36]. In this study, genes involved in carbohydrate metabolism, transcription factor-related, and stress response were found to be differentially expressed in response to salt stress. These genes have been reported to play a role in resisting or tolerating abiotic stress [37,38].

### 3.2. Effect of Salinity Stress on Carbohydrate and Energy Metabolism

The energy-consuming process of surviving under salt stress involves the utilization of carbohydrate metabolism, which produces soluble carbohydrates and a significant proportion of available ATP, to defend plants against stress-adaptive responses [39,40]. The expression of certain carbohydrate metabolism and energy genes were significantly altered in the salt-tolerant genotypes (‘Gongnong NO.1’) compared to the sensitive genotype (‘Sibeide’).

Carbohydrate metabolism plays a major role in osmotic regulation under salt stress [41]. UDP-glycosyltransferases (UGTs) are responsible for transferring sugar moieties onto various small molecules and play a crucial role in modulating metabolism and adapting to stress [42]. For instance, *Arabidopsis UGT_79_B2* and *UGT_79_B3* can enhance abiotic stress tolerance by modulating anthocyanin metabolism [43]. The rice glycosyltransferase gene *UGT2* plays a crucial role in salt stress tolerance [44]. In this study, we observed that a higher number of UDP glycosyltransferase genes were up-regulated in ‘Gongnong NO.1’ compared to the ‘Sibeide’ (Figure 5), suggesting that genes involved in carbohydrate metabolism may play crucial roles in salt tolerance in Alfalfa.

Salinity stress altered gaseous exchange, water availability, stomata closure, and photosynthesis. It also impacts electron flow in the chloroplast and mitochondria’s electron transport chain (ETC), leading to increased production of ROS and disturbed levels of adenine (ATP) and pyridine nucleotides (NADH, NADPH). Therefore, oxidative damage and altered energetics play a key role in impacting general metabolism and plant growth under salt stress [45]. Salt and osmotic stress can lead to overproduction of reactive oxygen species (ROS), which increases oxidative stress in plants [46]. Since these processes often involve a redox response, an additional NADPH supply may be necessary for all the pathways utilizing it [47]. In our study, we observed that NADP dehydrogenases were down-regulated at both the activity and protein/gene expression levels in ‘Sibeide’ (Appendix A). This finding further supports that ‘Sibeide’ is a salt-sensitive strain. AAA-type ATPases were first defined as ‘ATPases associated with diverse cellular activities’ [48], and they form a large and diverse superfamily found in all organisms. The maize AAA-type protein SKD1 was found to enhance salt tolerance and drought tolerance in transgenic tobacco by interacting with cleavage interaction protein 5 [49]. In our study, we observed changes in the expression levels of ATPase-related genes in ‘Gongnong NO.1’ and ‘Sibeide’ in response to salt stress. Our results demonstrate that under salinity stress, salt-sensitive alfalfa seedlings exhibit reduced carbohydrate utilization and energy production, leading to growth reduction and organic degradation (Figure 1). This finding suggests that the ‘Sibeide’ variety is more sensitive to salt than ‘Gongnong NO.1’. The presented data suggest that during early development stages of alfalfa seedlings under salt stress conditions, carbohydrate accumulation and energy generation are essential for cellular activity.

### 3.3. Role of Stress-Associated Family and TF-Encoding Genes in Salt-Stress

Salt stress-induced production of reactive oxygen species (ROS) disrupts redox homeostasis and causes oxidative damage to seedlings by accumulation H_2_O_2_, OH^−^, and O_2_^−^ [9,50]. The scavenging of H_2_O_2_, OH^−^, and O_2_^−^ during the stress response depends on enzymatic reaction catalyzed by antioxidant enzymes and non-enzymatic reaction composed of antioxidants such as peroxidase, and glutathione [51,52]. Transcriptome analysis showed consistent up-regulation of genes related to peroxidase and Glutathione S-transferase activity in ‘Gongnong NO.1’ but down-regulation in ‘Sibeide’ alfalfa under NaCl treatment (Appendix A). Therefore, it is speculated that plants under NaCl treatment might accumulate less H_2_O_2_ in ‘Gongnong NO.1’ than in ‘Sibeide’ alfalfa due to the activation of enzymatic and nonenzymatic reactions to scavenging endogenous H_2_O_2_ in the salt-tolerant species ‘Gongnong NO.1’.

Many transcriptome studies related to salt stress have been conducted in alfalfa, aiding our understanding of the molecular mechanism underlying salt tolerance. A diverse array of transcription factor (TF) families (e.g., NAC, WRKY, AP2/ERF, bZIP, and MYB) have been involved in regulating downstream genes responsible for salt tolerance in plants under salt stress. In this study, several important TF-encoding genes were significantly differently expressed under salt stress in ‘Gongnong NO.1’ or ‘Sibeide’ alfalfa seedings. These TFs belonged to the bHLH, WRKY, AP2/ERF, MYB, and NAC families, which are well known for their involvement in salt stress regulation. Most Zinc finger TFs were up-regulated in ‘Gongnong NO.1’ but down-regulated in ‘Sibeide’ after salt stress (Appendix A). ‘Gongnong NO.1’ alfalfa under salt stress has 12 genes encoding 4 types of zinc finger protein, with 6 of them up-regulated and 6 down-regulated, potentially playing an important role in salt adaption. Additionally, seven genes encode RING-type zinc finger proteins, four genes encode CCHC/CH-type zinc finger proteins and one gene (MsG0280010800) encodes RING-type zinc finger (up-regulated by aboutsix-fold). A total of 17 zinc finger proteins including RING-type, Dof-type, HD homeobox, basic-leucine zipper and zinc ribbon were differentially expressed. Among the 13 genes, most were down-regulated and 4 genes (MsG0580028462, MsG0580029912, MsG0780041106, and MsG0580027325) were up-regulated in ‘Sibeide’ under salt stress conditions. These zinc finger proteins could potentially act as positive regulatory factors in the salt stress response of two different varieties of alfalfa [53,54,55,56,57]. Two TFs (AP2/ERF and MYB) consistently exhibited higher expression levels in ‘Gongnong NO.1’ alfalfa under salt stress (Appendix A). CBF TFs were up-regulated, while three TFs (GRAS, Myb, and bHLH) showed lower expression levels in ‘Sibeide’ after salt stress (Appendix A). Numerous studies have shown that expression of abiotic stress-related genes, such as AP2/ERF, MYB4, bHLH, and bZIP, is associated with salt tolerance in plants [58,59,60]. This suggests that the difference in salt tolerance between ‘Gongnong NO.1’ and ‘Sibeide’ alfalfa may be attributed to varying expression levels of TFs. Further confirmation is required to elucidate the exact mechanisms of action of these TFs under salt stress in alfalfa.

### 3.4. Differentially Expressed Genes in Response to Salt-Stress

In our study, we observed significant differences in the expression levels of genes encoding stress-related protein. Appendix A show that 40 and 16 stress-related genes were identified in ‘Gongnong NO.1’ and ‘Sibeide’, respectively. These DEGs primarily encode proteins from the EF-hand family, heat shock proteins (HSPs), late embryogenesis abundant (LEAs), plant peroxidase, and glutathione S-transferases (GSTs). *HSPs* play a role in plant responses to high salt; *OsHsfA7*, *OsHsp17.0*, and *OsHsp23.7* play important roles in rice under salt and drought stresses [61,62]. Our findings indicate that DEGs encoding *HSPs* were up-regulated under salt stress. Similarly, plant GSTs have diverse roles in plant development, endogenous metabolism, and stress tolerance. For instance, the GST gene has the ability to enhance salt tolerance in transgenic *Arabidopsis* [63]. Our findings indicate that GST was up-regulated in the salt-resistant cultivar ‘Gongnong NO.1’ and down-regulated in the salt-sensitive cultivar ‘Sibeide’. This suggests that ‘Gongnong NO.1’ has a higher level of reduced glutathione to scavenge ROS. Peroxidase [64], EF-hand [65], and LEA [66] proteins have been shown to enhance plant resistance to abiotic stress. The observed differential expression of these stress-related genes highlights their significance in protecting Alfalfa from salt-stress damage.

### 3.5. MsHPCA1 Might Be a H_2_O_2_ Receptor in Alfalfa and Is Involved in Stress Responses

*Arabidopsis thaliana* HPCA1 is a membrane-localized leucine-rich repeat receptor-like kinase that functions as a central ROS receptor [67]. *HPCA1* is implicated in different physiological functions, including cell-to-cell ROS signals, systemic signaling and different stresses [34,68]. Additionally, *HPCA1* appears to be involved in water deficit and ABA signaling [69]. To further demonstrate the salt signaling pathway in alfalfa, we identified *MsHPCA1* as membrane-localized, indicating its putative role as a membrane receptor. The expression of *MsHPCA1* was significantly elevated in the salt-resistant ‘Gongnong NO.1’ but not in the salt-sensitive ‘Sibeide’. Furthermore, overexpression of *MsHPCA1* in ‘Sibeide’ increased salt tolerance. The expression of *MsHPCA1* was significantly elevated in both ‘Gongnong NO.1’ and ‘Sibeide’ after H_2_O_2_ treatment, and overexpression of *MsHPCA1* in ‘Sibeide’ increased H_2_O_2_ content. Overexpression of *MsHPCA1* in ‘Sibeide’ could enhance the salt tolerance of plants, while silencing of *MsHPCA1* in ‘Gongnong NO.1’ could reduce the salt tolerance of plants. The above data indicated that *MsHPCA1* might be involved in salt stress. By measuring the H_2_O_2_ content, we found that overexpression of *MsHPCA1* increased the accumulation of H_2_O_2_. To our knowledge, this is the second report on H_2_O_2_ receptors in plants, and in-depth physiological, transcriptional, and downstream signaling analyses would reveal its working mechanism.

## 4. Materials and Methods

### 4.1. Plant Materials Subjected to Salt Treatments

Vernalization: Alfalfa seeds (*Medicago sativa* L. ‘Gongnong No.1’ and ‘Sibeide’) were placed in a refrigerator at 4 °C for 24 h for vernalization. Disinfection and sterilization: approximately 10 g of alfalfa seeds were sterilized and then spread on sterilized filter paper on an ultra-clean table for ventilation.

Seed germination under salt stress conditions: two layers of sterilized qualitative filter paper were placed in each glass petri dish on an ultra-clean bench. Then, 4 mL of 250 mM NaCl solution were added, followed by the seeds. The plants were cultivated in a greenhouse under standard conditions, including a day/night temperature of 25 °C/20 °C ± 2 °C, 60% humidity, a photoperiod of 16 h light and 8 h dark, and a luminous flux density of 40 µmol/m^2^s. On day 7 after treatment, analysis of the germination rate and potential, seedling and root length, proline content, malondialdehyde (MDA) content, and chlorophyll content were measured.

### 4.2. Analysis of Physiological Parameters

Root length and seedling length: seedlings with similar growth were selected from each dish, and the length of both the root and shoot were measured. Germination rate: The germination rate was calculated daily starting for 7 days after seed germination. Germination was considered when the root broke through the seed coat to reveal a white bud. Germination rate = (number of germinations on the seven days/number of seeds tested) × 100%. Germination potential: germination potential was calculated as the number of germinations on the fourth, sixth, and eighth days divided by the number of seeds tested, multiplied by 100%. Proline content and MDA content were measured using detection kits (Suzhou Keming Biotechnology Co., Ltd., Suzhou, China) following the manufacturer’s instructions. Chlorophyll content: The chlorophyll content of plant leaves was measured using a portable chlorophyll meter (Model Specifications: SPAD-502PLUS, Konica Minolta, Tokyo, Japan).

### 4.3. Analysis of RNA-Seq Data, Assembly of Sequences, and Annotation of Genes

Seeds of alfalfa varieties ‘Gongnong No.1’ and ‘Sibeide’ were germinated for 7 days and then exposed to 250 mM NaCl for durations of 0, 12, and 24 h. The whole seedlings of three biological replicates were individually collected at each treatment time. All samples were immediately frozen in liquid N_2_ and stored at −80 °C until the RNA extraction. The seedlings collected from each treatment were used for total RNA isolation using the TRIzol Total RNA Extraction Kit (Kangwei Century Biotechnology Co., Ltd., Beijing, China), and cDNA library preparation and sequencing reactions were performed at the Novogen Bioinformatics Institute (Novogen, Beijing, China). The raw reads were processed using Trimmomatic [70]. A threshold of both a *p*-value < 0.05 and log2 FC > 2 was used to identify genes with significantly different levels of expression. GO and KEGG pathway enrichment analyses of all DEGs detected during different periods of salt were performed in R using the hypergeometric distribution. The reads were reassembled using StringTie [71].

### 4.4. Identification of Differentially Expressed Genes

We identified the DEGs of two alfalfa varieties, “Gongnong NO.1” and “Sibeide”. Subsequently, we utilized the AlfalfaGEDB (http://alfalfagedb.liu-lab.com/blastplus/blast, accessed on 26 June 2023), Medical Analysis Portal (https://medicago.legumeinfo.org/, accessed on 26 June 2023), UniProtKB1 (https://www.uniprot.org/uniprotkb, accessed on 23 August 2023) and the TAIR2 (https://www.arabidopsis.org/, accessed on 1 March 2024) as a reference to identify and annotate the genes that are DEGs in this experiment. Simultaneously, we utilized the AlfalfaGEDB and Medicago Analysis Portal databases to identify genes with significantly associated loci, which were related to the DEGs that had undergone preliminarily screening. We refer to the SNPs located on eight chromosomes of alfalfa [72], which are associated with salt stress. Then, the findings from the RNA-seq analysis in this study were integrated to analyze candidate genes with salt stress tolerance in alfalfa.

### 4.5. Quantitative Real-Time PCR Analysis

Total RNA was isolated from various seedling samples using the TRIzol reagent (Invitrogen, Waltham, MA, USA) according to the manufacturer’s protocol. cDNA was generated from 2 μg of DNA-free RNA using the RevertAid First Strand cDNA Synthesis Kit (Fermentas, Glen Burnie, MD, USA) as per the manufacturer’s instructions. For RT-qPCR experiments, three biological replicates were conducted to ensure reproducibility and statistical significance. The relative expression levels were determined using the 2^−ΔΔCt^ method. The *MsActin* gene was used as a house-keeping gene in RT-qPCR. The gene-specific primers used are provided in Appendix A.

### 4.6. Statistical Analysis

The experimental data were organized using Microsoft Excel. Mean, standard deviation, and standard error were computed. A two-way ANOVA analysis was conducted using GraphPad Prism 9, a statistical analysis software package. Significant differences were determined using the Duncan multiple-range test. The seed germination rate, germination potential, root length, and seedling length were determined by the average measurements from three replicates. The physiological indicators were assessed by averaging measurements from six replicates. Transcriptome sequencing was conducted using three biological replicates.

### 4.7. Development of HPCA1 Overexpression and RNAi Alfalfa Lines

Based on the recent study conducted by Wang et al. [15], we have successfully transferred the pKGWRR-OE-*MsHPCA1* and pKGWRR-tasi-*MsHPCA1* into the alfalfa varieties ‘Gongnong No.1’ and ‘Sibeide’ using *Agrobacterium rhizogenes*-mediated hairy root transformation. After sterilization, the alfalfa seeds were evenly distributed and cultured in a 0.8% Flame Spectroscopy (FS) solid medium. Once the root length reaches approximately 1 cm, it is trimmed using a sterile knife to remove approximately 3 mm of the root tip. The wound is then immersed in an *Agrobacterium tumefaciens* solution. Subsequently, it is transferred to a 1.5% FS solid medium that has been sterilized and incubated in a constant temperature light incubator and cultured for 3 weeks. We utilized a handheld dual-wavelength fluorescent lamp (LUYOR-3415RG, Luyor Corporation, Shanghai, China) to excite root RFP signals at 520 nm, aiming to verify the transgenic seedlings, RT-qPCR was used to detect the expression of *MsHPCA1* in transgenic plants.

The positive plants were transferred to a container containing a mixture of soil and vermiculite (in a 1:1 ratio) and were subsequently cultivated in a greenhouse for one week. After subjecting the seedlings to 10 days of irrigation with 250 mM NaCl, we evaluated various physiological and biochemical indicators, including seedling length, root length, plant height, and chlorophyll content in leaves, and the endogenous hydrogen peroxide content (Suzhou Keming Biotechnology Co., Ltd., Suzhou, China).

### 4.8. The Relative Expression Level of MsHPCA1

Then, we investigated the expression pattern of *MsHPCA1* gene in alfalfa. We used RT-qPCR to evaluate the relative expression levels of *MsHPCA1* in diverse tissues, including roots, stems, leaves, and flowers.

In addition, after a 3-week cultivation period of two alfalfa varieties, ‘Gongnong No.1’ and ‘Sibeide’, various treatments were applied to the seedlings. These included drought stress, application of 250 mM NaCl and 100 mM H_2_O_2_ (hydrogen peroxide). After 10 days of various treatments, the relative expression level of *MsHPCA1* was determined and evaluated in the treated seedlings.

### 4.9. The Subcellular Localization of MsHPCA1

For the localization study, the ORF sequence of *MsHPCA1* was amplified and cloned into the vector pEarleyGate103-SL. The recombinant plasmid was then introduced into *Agrobacterium* (GV3101), and it was used to infect the leaves of *Nicotiana benthamiana* after 36 h. The GFP fluorescence (excitation at 488 nm) and the mCherry fluorescence from the membrane marker protein PTIG6 (excitation at 580 nm) were excited using a confocal laser scanning microscope (Nikon AX, Tokyo, Japan) for the same experiment.

### 4.10. Accession Numbers

*Arabidopsis thaliana AtHPCA1* sequence (Accession No. AT5G49760) and *MsActin* (Accession No. JQ028730.1) sequences were obtained from the NCBI database. A putative *MsHPCA1* (Unigene ID number: MsG0280007602.01) was identified from transcriptome data of *Medicago sativa* L. via homology search using the *Arabidopsis thaliana AtHPCA1* sequence.

### 4.11. Bioinformatics Analysis of MsHPCA1 Gene

*MsHPCA1* sequences were retrieved from *Medicago sativa* transcriptome data. All other sequences were identified from a public database, such as NCBI. The protein sequences were multiple compared using clustalW. MEGA 7.0 is used to build the phylogenetic tree through the maximum likelihood method, and the bootstrap method is used to verify the quality of tree building, with 1000 times of inspection. Conserved motifs of MsHPCA1 proteins were predicted using the SMART (http://smart.embl-heidelberg.de/, accessed on 16 December 2023) with default parameters. The coding sequences (CDS) and the structure of all genes were graphically displayed with TBtools function “Gene Structure View”. The three-dimensional structure of protein was predicted using SWISS-MODEL (https://swissmodel.expasy.org/, accessed on 16 December 2023) with a confidence level of 0.84.

## 5. Conclusions

By analyzing the transcriptome of the ‘Gongnong NO.1’ and ‘Sibeide’ alfalfa under salt treatment, we identified a series of genes that were significantly induced to generate adaptive responses, thereby alleviating the damage caused by salt stress. These genes are involved in glyco-metabolism, plant energy, and stress-associated and transcription factors. We found that *MsHPCA1* was specially induced in the salt-tolerant alfalfa cultivar (‘Gongnong NO.1’) under salinity stress, but not in the salt-sensitive cultivar (‘Sibeide’). We further functionally characterized the *MsHPCA1* by transforming *MsHPCA1* to alfalfa, and found that transgenic alfalfa overexpressing *MsHPCA1* displayed enhanced salt stress tolerance and antioxidant activities. Collectively, this research offers new insight into the molecular mechanism of alfalfa salt stress tolerance and highlights the potential utility of *MsHPCA1* in alfalfa breeding.

## Figures and Tables

**Figure 1 plants-13-01073-f001:**
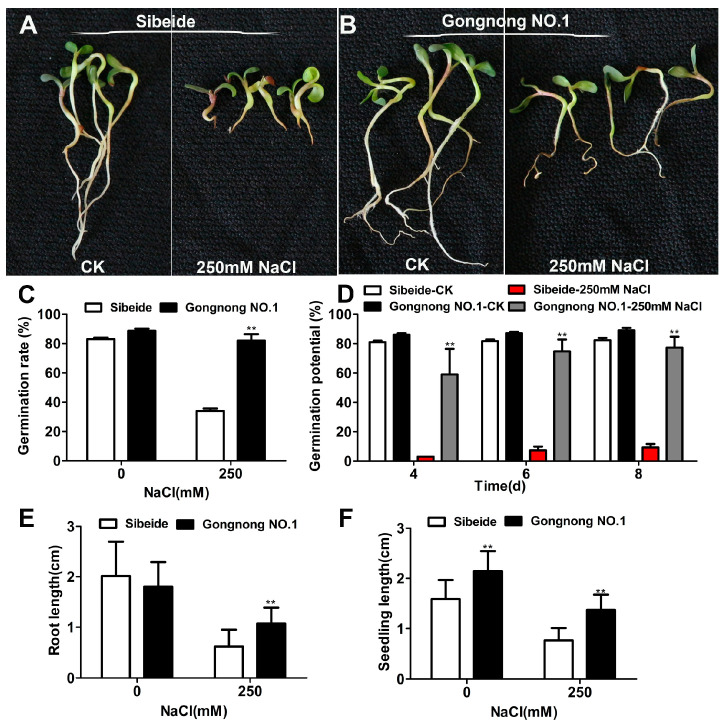
Salt stress inhibited the germination and growth of different alfalfa varieties. (**A**,**B**) Phenotype of ‘Sibeide’ and ‘Gongnong NO.1’ before and after salt treatment. (**C**) Germination rate. (**D**) The germination potential of ‘Gongnong NO.1’ and ‘Sibeide’, changed at 4, 6, and 8 days of salt treatment. (**E**) Root length. (**F**) Seedling length. After 7 days of 250 mM NaCl treatment, the germination rate, germination potential, root length, and seedling length were counted. Germination rate and germination potential were tested with three experiments (*N* = 100). Root length and seedling length were also subjected to three experiments (*N* = 10). The plot represents the mean ± standard deviation. The asterisk indicates a significant difference between ‘Sibeide’ and ‘Gongnong NO.1’ salts before and after treatment. Analysis was performed using the *t*-test. ** indicates *p* < 0.01. CK: Water, NaCl: 250 mM.

**Figure 2 plants-13-01073-f002:**
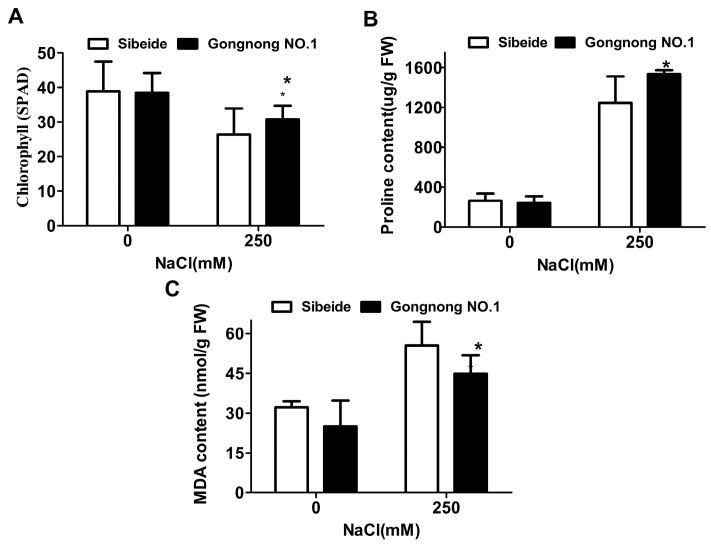
‘Sibeide’ is more sensitive to salt than ‘Gongnong NO.1’. (**A**) Chlorophyll content. (**B**) Proline content. (**C**) MDA content. The changes of physiological indexes were detected after 7 days of treatment with 250 mM NaCl for ‘Sibeide’ and ‘Gongnong NO.1’. Analysis was performed using the *t*-test, three replicate experiments were performed (*N* = 10), and error bars indicate ± SE of means. The asterisk indicates a significant difference between ‘Sibeide’ and ‘Gongnong NO.1’ salts before and after treatment (*t*-test, * *p* < 0.05). 0: Water, 250: 250 mM NaCl.

**Figure 3 plants-13-01073-f003:**
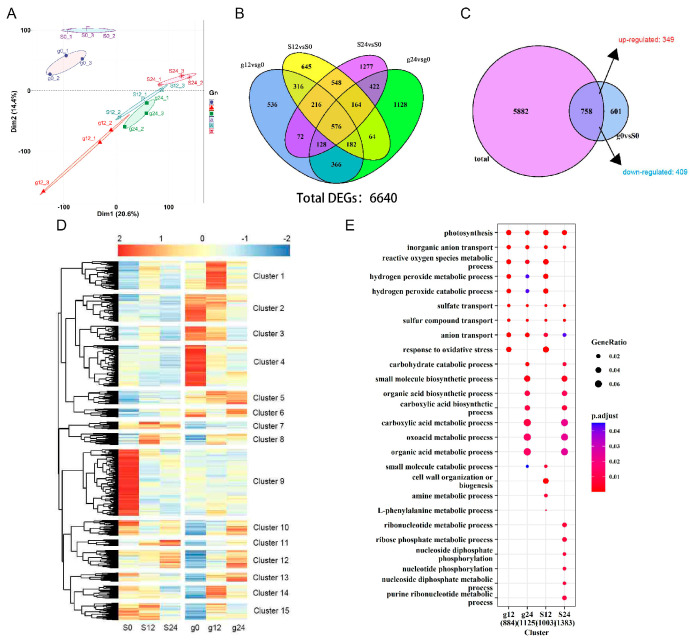
DEGs were compared between salt stress treatment at 12 h and 24 h with 0 h. (**A**) The PCA analysis indicated the DEGs of g0, g12, g24, S0, S12, and S24. (**B**) The Venn diagram illustrates the number of statistically significant DEGs in the transcriptome data for the comparison between g12 vs. g0, g12 vs. g0, S12 vs. S0, and S24 vs. S0, where ‘g’ represents ‘Gongnong NO.1′and ‘S’ represents ‘Sibeide’. (**C**) The Venn diagram illustrates the intersection of co-expressed DEGs between ‘Gongnong NO.1’ (12 h, 24 h) and ‘Sibeide’ (12 h, 24 h) with the DEGs of the two varieties at 0 h. (**D**) A heatmap illustrates the relative expression levels of DEGs in ‘Gongnong NO.1’ (0 h, 12 h, and 24 h), and ‘Sibeide’ (0 h, 12 h, and 24 h). (**E**) Scatter plots display KEGG enrichment analysis of DEGs in ‘Gongnong NO.1’ (12 h, 24 h) and ‘Sibeide’ (12 h, 24 h) compared to their respective controls at 0 h. Each color represents a gene set of unigenes, that are co-expressed or unique among gene sets. The sum of all numbers within each circle represents the total number of unigenes in the set, and the overlapping areas between circles represent the number of co-expressed unigenes among the gene sets.

**Figure 4 plants-13-01073-f004:**
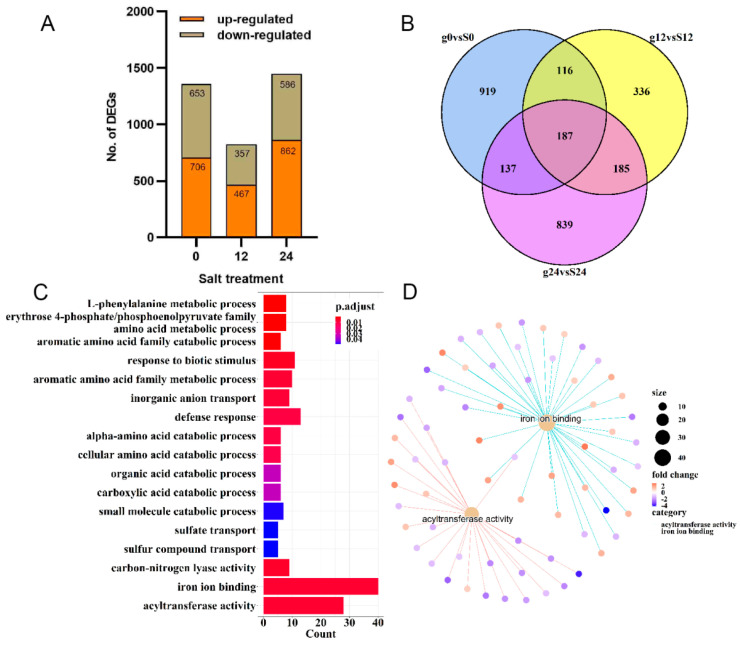
Comparison of differentially expressed genes in g0 vs. S0, g12 vs. S12, and g24 vs. S24. (**A**) Distribution of the sets of up- and down-regulated genes in ‘Gongnong NO.1’ and ‘Sibeide’ under salt treatment for 0 h, 12 h, and 24 h. The brown box above represents the down-regulated DEGs, and orange represents the up-regulated. (**B**) Venn diagram presenting g0 vs. S0, g12 vs. S12, and g24 vs. S24 co-expressed DEGs genes when exposed to NaCl stress conditions. (**C**) KEGG pathway enrichment analysis of co-expressed DEGs. (**D**) Co-expressed DEG mainly enriched in iron ion binding and acyltransferase activity.

**Figure 5 plants-13-01073-f005:**
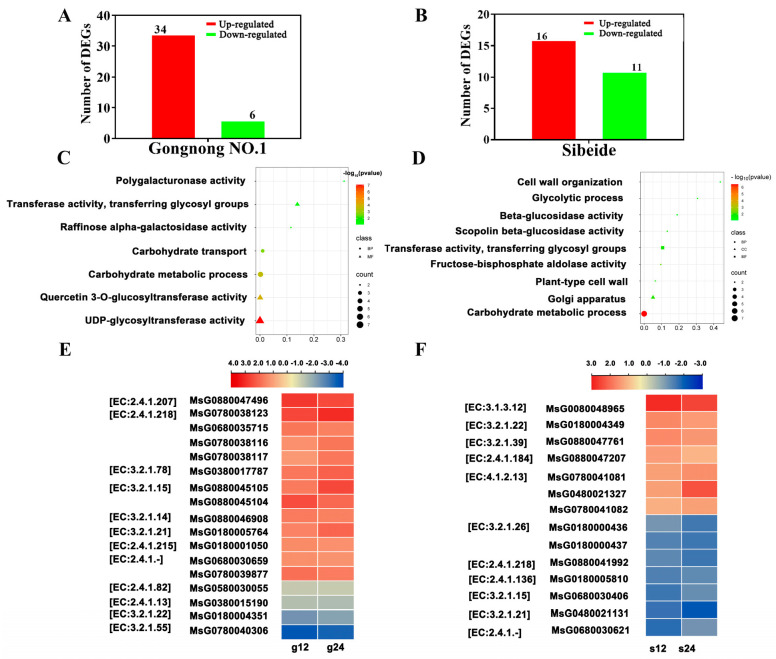
Expression analysis of glyco-metabolism-associated genes in ‘Gongnong NO.1’ and ‘Sibeide’. (**A**,**B**) The number of genes related to glucose metabolism were up- and down-regulated in ‘Gongnong NO.1’ (12 h, 24 h) and ‘Sibeide’ (12 h, 24 h). (**C**,**D**) GO analysis of genes associated with glucose metabolism in ‘Gongnong NO.1’ and ‘Sibeide’. The color of each circle represents the GO functional classification (BP: Biological Process, MF: Molecular Function, CC: Cellular Component), the size indicating the number of differential proteins within each function. The *x*-axis indicates *p*-values of enrichment significance and the *y*-axis is named for the GO function. (**E**,**F**) The expression of the genetically encoded enzyme changed after 12 h and 24 h of salt treatment. The red and blue represent up- and down-regulation, respectively. Enzymes are given as EC numbers.

**Figure 6 plants-13-01073-f006:**
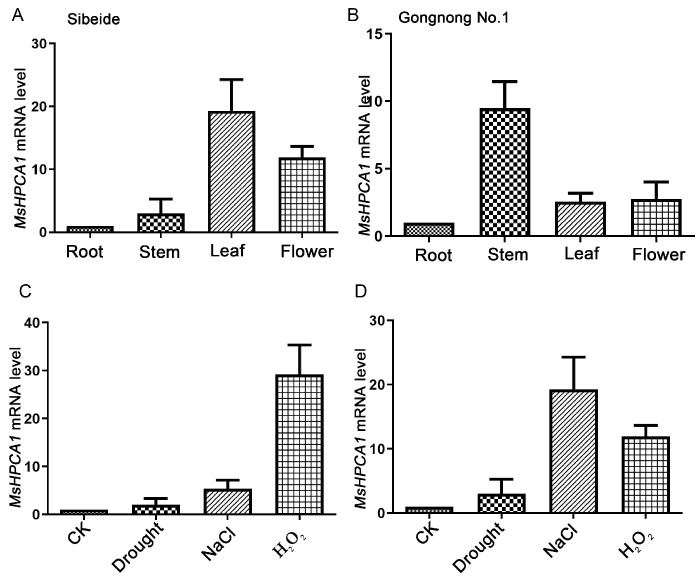
*MsHPCA1* was identified as a salt-responsive gene. (**A**,**B**) Tissue-specific expression patterns of the *MsHPCA1* gene in ‘Gongnong NO.1’ and ‘Sibeide’. (**C**,**D**) The relative expression level of *MsHPCA1* in leaves after 10 days of exposure to drought stress, 250 mM NaCl and 100 mM H_2_O_2_. The plot represents the mean ± SD of three repeats. CK: control, Drought: stop watering, NaCl: 250 mM NaCl, H_2_O_2_: 100 mM H_2_O_2_.

**Figure 7 plants-13-01073-f007:**
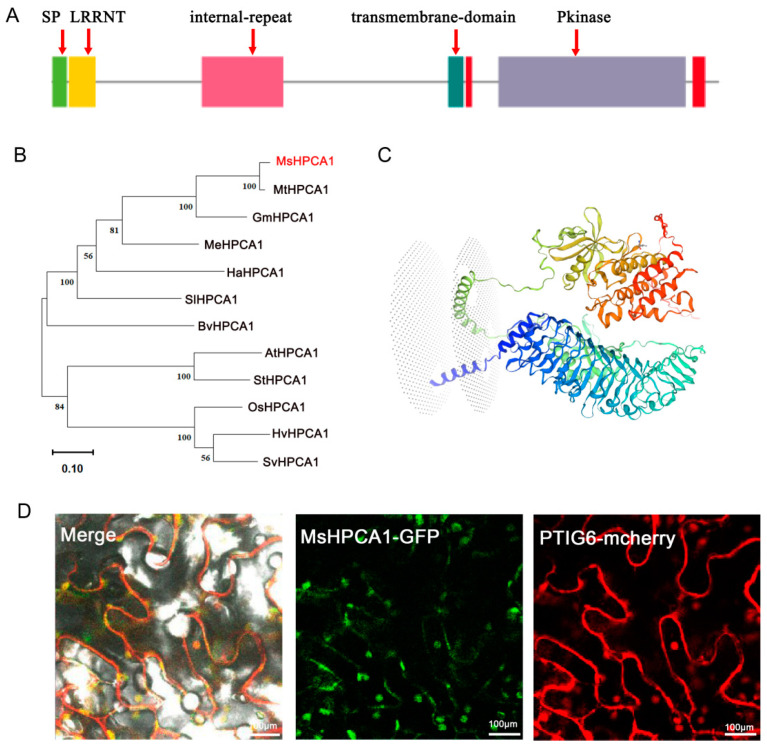
Bioinformatics analysis and subcellular localization of MsHPCA1. (**A**) Predicted domain architecture of MsHPCA1 proteins. SP: signal peptide. Each distinct structural domain is represented by a colored rectangle. (**B**) Phylogenetic analysis of HPCA1 homologs in different plant species. MsHPCA1: *Medicago sruncatula*, MsG0280007602.01; BvHPCA1: *Beta vulgaris*, XP_019105897.2; OsHPCA1: *Oryza sativa*, XP_015619793.1; MeHPCA1: *Manihot esculenta*, XP_021592399.1; MtHPCA1: *Medicago truncatula*, XP_003594434.2; GmHPCA1: *Glycine max*, XP_025979435.1; SvHPCA1: *Setaria viridis*, XP_034588472.1; AtHPCA1: *Arabidopsis thaliana*, NP_001321900.1; StHPCA1: *Solanum tuberosum*, XP_006338928.1; SlHPCA1: *Solanum lycopersicum*, XP_004230391.1; HaHPCA1: *Helianthus annuus*, XP_021981660.1; HvHPCA1: *Hordeum vulgare*, XP_044974898.1. (**C**) Three-dimensional structure of MsHPCA1 proteinThe red characters represent HPCA1 alfalfa, whereas the black letters represent other species. (**D**) Subcellular localization of MsHPCA1, PTIG6-mcherry as a marker for membranes.

**Figure 8 plants-13-01073-f008:**
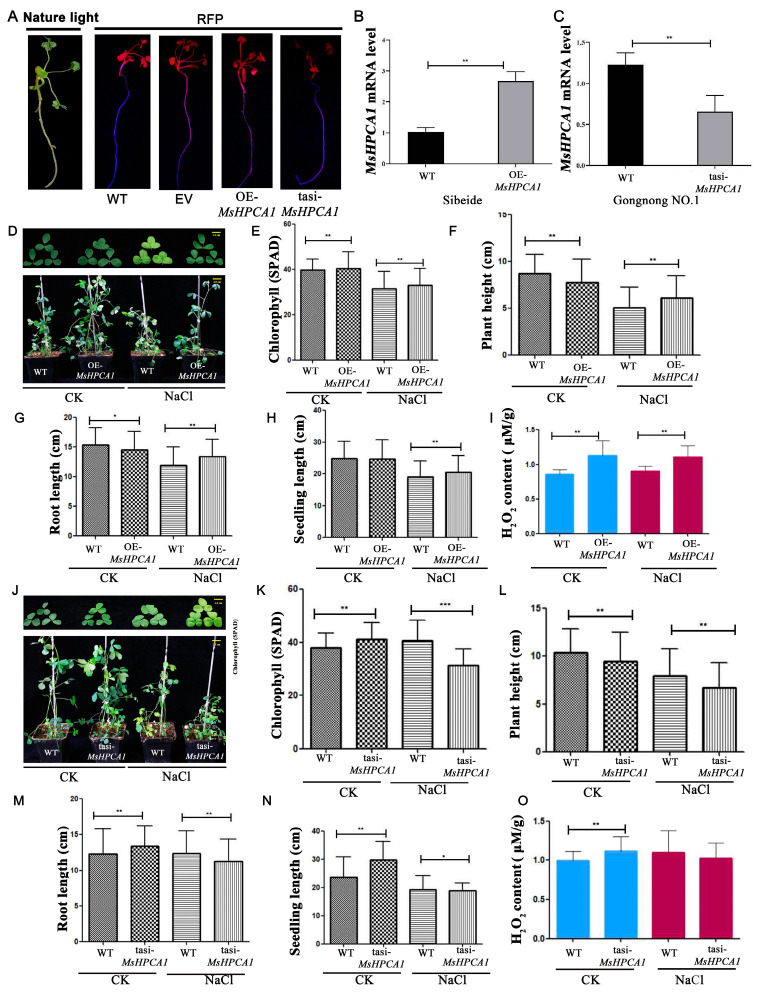
Phenotypes and physiological change of alfalfa with tasiRNA-silenced and overexpression of *MsHPCA1* gene under salt stress. (**A**) Confirmation of hairy root transformation by red fluorescent protein (RFP) expression. The left panel is the alfalfa hair root taken under natural light, and the right panel is the alfalfa hair root photographed under a wavelength fluorescent lamp. (**B**,**C**) RT-PCR was used to detect the overexpression of ‘Sibeide’ and the silencing efficiency of ‘Gongnong NO.1’. (**D**,**J**) The phenotypes of *MsHPCA1*-transgenic and WT plants under salt treatment. (**E**,**K**) Chlorophyll content. (**F**,**L**) Plant height. (**G**,**M**) Root length. (**H**,**N**) Seedling length. (**I**,**O**) H_2_O_2_ content. tasi, *MsHPCA1*-silenced plants; OE, *MsHPCA1*-overexpression plants. CK: irrigated with water for 10 days, NaCl: irrigated with 250 mM NaCl solution for 10 days. The photographs were taken, and the root and seedling length, plant height, and chlorophyll of each plant were measured. Data are represented as the mean ± SD (*N* = 5). The asterisk indicates a significant difference between transgenic lines and WT plants (*t*-test, * *p* <0.05, ** *p* < 0.01, *** *p* < 0.001).

## Data Availability

Data are contained within the article and Appendix A. RNA sequencing data has been uploaded to the NCBI Sequence Read Archive under accession ID PRJNA1094402.

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
