# Peer review of "Transcriptome Analysis for Salt-Responsive Genes in Two Different Alfalfa (Medicago sativa L.) Cultivars and Functional Analysis of MsHPCA1"

_plants, 2024, doi:10.3390/plants13081073_

Round 1

Reviewer 1 Report

Comments and Suggestions for Authors

The article entitled " Transcriptome analysis for salt-responsive genes in two different alfalfa (Medicago sativa L.) cultivars and functional  analysis of MsHPCA1 " has tried to identify salt-responsive gene and functional  analysis of MsHPCA1 in alfalfa.  Some issues need to be concerned.

1. All the figures can't see clearly and some of them were very small. The figures should be high resolution.

2. Why the authors choose the MsHPCA1 gene for further function analysis? Did the MsHPCA1 gene include in the DEGs and which bioprocess was involed in ? MsHPCA1 was stress-associated gene? 

3. The functional validation of MsHPCA1 is not very relevant with the transcriptome analysis. These two parts may be independent.The authors should be establish the connection between them carefully with effective evidence.Many times,the corresponding figures or tables are needed.

4. How about the expression patterns of MsHPCA1in the two cultivars by transcriptome analysis ? Figure 6(C, D) Expression analysis of MsHPCA1 under different stress treatments. Whether these are from different cultivars and what do they correspond to cultivars?

5. Figure 8. Phenotypes and physiological change of alfalfa with knockout and overexpression of MsHPCA1 gene under salt stress. Here the "knockout" is inappropriate, just tasiRNA-silenced.

6. In Mehtods, 4.4 "Based on the linkage disequilibrium (LD) of the  association panel, we identified all genes within 200 kb (100 kb up- and downstream) of the significant loci. The genes located near the significantly associated loci have garnered increased attention." How the authors used the LD of the association panel? And what did the significant loci?

7. The transcriptome sequencing should be deposited in NCBI or National Genomics Data Center.

Comments on the Quality of English Language

Moderate editing of English language required.

Reviewer 2 Report

Comments and Suggestions for Authors

The authors performed a transcriptome analysis of two alfalfa varieties, salt-tolerant variety Gongnong NO.1 and salt-sensitive variety Sibeide, under salt stress and identified many DEGs involved in carbohydrate metabolism, energy production, transcription factor, and stress-associated pathway. Further, they analyzed the function of MsHPCA1 in salt response of alfalfa by utilizing OE and RNAi lines. Although the data provided support their overall conclusion, many critical issues should be considered to support the conclusion and clarify the manuscript.

Figure 1; the method of statistical analysis should be described in figure legend: means and SE, number of replication, p-value, etc.

Figure 2; it is described in page 3 that MDA content was lower in 'Gongnong NO.1' than in 'Sibeide' under salt stress, which is wrong. Figure 2C shows that MDA content is higher in Gongnong NO. 1 than Sibeide. The y-axis on Fig. 2C should be MDA.

Figure 5; the figure legend “the red and blue arrows represent upregulation and downregulation” is wrong: it should be the red and blue colors represent…

Figure 6; the method of statistical analysis should be described in figure legend: number of replication, p-value, etc.

Figure 7; the description of Fig. 7 in page 8 under the subtitle “subcellular localization of MsHPCA1” is misleading: Fig. 7B was wrongly quoted, and Fig. 7C was never mentioned in the text. In Fig. 7D legend, PTIG6-mcherry should be explained: is it a membrane marker?

Figure 8; to confirm the function of a gene using transgenic approach, multiple transgenic lines (at least three lines) should be analyzed. In addition, the levels of MsHPCA1 in the OE and RNAi lines should be analyzed by RT-PCR to confirm OE and RNAi lines. Moreover, the response of the OE and RNAs lines to salt stress should be analyzed in detail; for instance, root length, seedling length, fresh weight, survival rate, etc., should be measured. It is described in page 9 that MsHPCA1-overexpression plant accumulated more H2O2 compared with WT plant, indicating that MsHPCA1-mediated salt tolerance might be associated with H2O2 accumulation (Figure 8D, E). This description cannot be supported from the current data. Fig. 8D shows that H2O2 contents in the WT and OE line are similar between CK and NaCl, which indicates that overexpression of MsHPCA1 does not affect H2O2 levels under salt conditions. Moreover, it is hard to understand that the salt-tolerant phenotype of the OE line is correlated with high H2O2 content. In general, salt tolerance of plants is correlated with less H2O2 levels.

Many DEGs, including EF-hand family genes, heat shock proteins, Late embryogenesis abundant, plant peroxidase, and glutathione S-transferases, were identified in salt-tolerant cultivar from RNA-seq analysis but HPCA1 was selected for functional analysis. Why? Is it the most extensively expressed gene upon salt treatment?

All abbreviations should be defined at its first appearance; for instance, PCA, KEGG in page 4.

Comments on the Quality of English Language

Some parts of the manuscript need English correction.

Round 2

Reviewer 2 Report

Comments and Suggestions for Authors

As the authors addressed all of my comments and suggestions, I have no further concerns.